## [Decision Letter · Decision Letter 0]

1 Aug 2025

Dear Dr. Herrera,

Thank you for submitting your manuscript to PLOS ONE. After careful consideration, we feel that it has merit but does not fully meet PLOS ONE’s publication criteria as it currently stands. Therefore, we invite you to submit a revised version of the manuscript that addresses the points raised during the review process.

We look forward to receiving your revised manuscript.

Kind regards,

Barathan Balaji Prasath

Academic Editor

PLOS ONE

Journal Requirements:

3. http://creativecommons.org/licenses/by/4.0/).%20Please%20be%20aware%20that%20this%20license%20allows%20unrestricted%20use%20and%20distribution,%20even%20commercially,%20by%20third%20parties.%20Please%20reply%20and%20provide%20explicit%20written%20permission%20to%20publish%20XXX%20under%20a%20CC%20BY%20license%20and%20complete%20the%20attached%20form.”%0b%0bPlease%20upload%20the%20completed%20Content%20Permission%20Form%20or%20other%20proof%20of%20granted%20permissions%20as%20an%20%22Other%22%20file%20with%20your%20submission.%0b%0bIn%20the%20figure%20caption%20of%20the%20copyrighted%20figure,%20please%20include%20the%20following%20text:%20“Reprinted%20from%20%5bref%5d%20under%20a%20CC%20BY%20license,%20with%20permission%20from%20%5bname%20of%20publisher%5d,%20original%20copyright%20%5boriginal%20copyright%20year%5d.”%0b%0bb.%20If%20you%20are%20unable" http://creativecommons.org/licenses/by/4.0/).%20Please%20be%20aware%20that%20this%20license%20allows%20unrestricted%20use%20and%20distribution,%20even%20commercially,%20by%20third%20parties.%20Please%20reply%20and%20provide%20explicit%20written%20permission%20to%20publish%20XXX%20under%20a%20CC%20BY%20license%20and%20complete%20the%20attached%20form.”%0b%0bPlease%20upload%20the%20completed%20Content%20Permission%20Form%20or%20other%20proof%20of%20granted%20permissions%20as%20an%20%22Other%22%20file%20with%20your%20submission.%0b%0bIn%20the%20figure%20caption%20of%20the%20copyrighted%20figure,%20please%20include%20the%20following%20text:%20“Reprinted%20from%20%5bref%5d%20under%20a%20CC%20BY%20license,%20with%20permission%20from%20%5bname%20of%20publisher%5d,%20original%20copyright%20%5boriginal%20copyright%20year%5d.”%0b%0bb.%20If%20you%20are%20unable" We note that Figure(s) 1 and 8 in your submission contain [map/satellite] images which may be copyrighted. All PLOS content is published under the Creative Commons Attribution License (CC BY 4.0), which means that the manuscript, images, and Supporting Information files will be freely available online, and any third party is permitted to access, download, copy, distribute, and use these materials in any way, even commercially, with proper attribution. For these reasons, we cannot publish previously copyrighted maps or satellite images created using proprietary data, such as Google software (Google Maps, Street View, and Earth). For more information, see our copyright guidelines: http://journals.plos.org/plosone/s/licenses-and-copyright.

a. You may seek permission from the original copyright holder of Figure(s) 1 and 8 to publish the content specifically under the CC BY 4.0 license.

4. Please remove all personal information, ensure that the data shared are in accordance with participant consent, and re-upload a fully anonymized data set.

Reviewers' comments:

Reviewer's Responses to Questions

**Comments to the Author**

1. Is the manuscript technically sound, and do the data support the conclusions?

Reviewer #1: Partly

Reviewer #2: Yes

2. Has the statistical analysis been performed appropriately and rigorously?

Reviewer #1: Yes

Reviewer #2: I Don't Know

3. Have the authors made all data underlying the findings in their manuscript fully available?

Reviewer #1: Yes

Reviewer #2: Yes

4. Is the manuscript presented in an intelligible fashion and written in standard English?

Reviewer #1: No

Reviewer #2: No

Reviewer #1: Abstract: The abstract is well-written, but in line 19, a comma should be placed after variables, and in line 23, it is recommended to add it before to.

Introduction: The introduction is concise and to the point, avoiding unnecessary details. It only contains some typographical and grammatical errors, all of which have been highlighted in different colors.

In this paragraph (Lines 49 to 50), there were several grammatical errors, which are outlined below.

• It is better to place a comma between coordinate adjectives. Therefore, in "Gram-negative photosynthetic prokaryotes," a comma should be added: "Gram-negative, photosynthetic prokaryotes"

• After "so", a complete sentence must follow. Therefore, the subject should be repeated after it, and the word "they" needs to be included.

• The phrase "extreme levels of biomass" sounds unnatural. In scientific and more natural language, it is better to say: "extremely high biomass levels".

• In line 53, "environment concentrations" should be corrected to "environmental concentrations." The reason it is incorrect is that "environment" is a noun, but here an adjective is needed to describe the noun "concentrations." The correct adjective form is "environmental."

• In line 65, the comma after "bacteria and" should be removed, but a comma after "cyanobacteria" is necessary to correctly separate the phrase.

Materials and methods: The authors have made an effort to explain the methodology completely; however, some parts remain unclear and require rewriting and further clarification. Additionally, there are several grammatical and stylistic issues in this section that have been highlighted.

• In line 88, the term "minimum level" is incorrect because the unit km³ refers to volume, not to the water level or elevation. It should be corrected to "minimum volume".

• This may raise questions for the reader about why the initial maximum volume differs from the current maximum volume (since the current maximum volume is stated as 0.529 km³). It would have been better to explain that the initial volume refers to the design or post-construction capacity, which has later decreased due to factors such as sedimentation.

• Although the six sampling sites (E1–E6) appear to represent hydrologically or anthropogenically distinct zones of the reservoir, the manuscript lacks a clear scientific or operational rationale for their selection. Please provide detailed reasoning, such as hydrodynamic characteristics, proximity to pollution sources, or accessibility criteria, that justifies the choice of these specific locations.

• The sentence: "Additionally, at each point, a drag sample was obtained with a phytoplankton net (Biológika, Colombia) with a pore diameter of 25 μm at a speed of approximately 3 m/s." contains a stylistic issue due to the repeated use of the preposition "with" in close succession. This repetition weakens the sentence structure and reduces readability. The Suggested revision: "Additionally, at each site, a drag sample was obtained using a phytoplankton net (Biológika, Colombia) featuring a pore diameter of 25 μm, towed at approximately 3 m/s."

• The phrase "This process consisted of filtering water for 5 minutes to concentrate the surface phytoplankton and perform a qualitative analysis of cyanobacteria and microalgae..." contains a grammatical inconsistency. After the phrase "consisted of," parallel structure should be maintained, meaning that both actions should be in the gerund (-ing) form.

• The sentence: "The collected material was analyzed directly under a microscope, and another part was fixed in-situ with Lugol" contains two parts, and it's better to connect them with while. "The Suggested revision: The collected material was partly analyzed directly under a microscope, while the remaining portion was fixed in situ with Lugol’s solution."

• Two types of sampling were conducted, each with distinct objectives, which explains the differences in sample volume and collection methods. The lower-volume samples were collected from various depths within the reservoir at points E1 to E6, across four time periods and three depth levels (totaling 72 samples), to assess the natural density of cyanobacteria. In contrast, the higher-volume samples collected from the tributaries were intended for growth experiments. However, the timing and frequency of the tributary sampling were not specified. It would have been better if the authors had provided this information to allow for a more accurate temporal comparison and a clearer analysis of the relationship between the inflows and the reservoir’s condition. The absence of these details introduces ambiguity in the interpretation of the data.

• It would have been better if the authors had indicated the locations of the four tributaries containing wastewater (T1 to T4) on the reservoir map or in Table 1. Displaying the geographical positions of these inflows is essential for a better understanding of the role of pollution sources in the development of cyanobacterial blooms. The absence of this information makes the spatial interpretation of the data and results more difficult.

• The phrase "mid-depth (50% of the photic zone limit)" may cause some ambiguity. It would be better to clarify precisely what is meant by the "photic zone limit" (for example, the depth at which 1% of surface light penetrates) and explain how the depths were determined.

• The manuscript does not provide any explanation for the selection of the four sampling months (February, April, July, and October). It would be beneficial for the authors to clarify whether these time points were chosen based on seasonal patterns, historical records of cyanobacterial blooms, hydrological events, or logistical constraints. Providing this information would strengthen the scientific rationale behind the sampling design. Furthermore, the time intervals between the sampling campaigns are inconsistent: there is a one-month gap between the first and second campaigns, a two-month gap between the second and third, and again a two-month gap between the third and fourth. It is important to explain the reasoning behind this irregular timing. What was the underlying logic for choosing these particular intervals?

• The comma after the word "visualization" is unnecessary and should be removed.

• Reference number 8 has been incorrectly formatted in italics and should be corrected.

• The term “acid preparations” is not clearly defined: From a technical standpoint, it is unclear what specific method is being referred to. Does this simply mean the use of India ink, or were actual acid-based preparations employed (e.g., to remove calcium carbonate or to break down cell walls)? If a particular acid treatment was used, it is important to specify the type of acid, concentration, duration of exposure, and the intended purpose of the treatment.

• The identification and enumeration of Microcystis spp. also raises questions: Was the identification based solely on morphology? Since Microcystis species often require genetic markers or fluorescence-based methods for reliable identification, the manuscript should acknowledge the limitations of using only light microscopy if that was the sole method.

• There is no mention of replication in the cell counting process: For methodological rigor, especially in quantitative studies, it is standard to perform counts in at least triplicate to minimize human error and improve reproducibility.

• Furthermore, the sample processing steps before microscopy are not described: It would be helpful to clarify whether any pre-treatment steps such as centrifugation, filtration, or dilution were performed before microscopic observation.

• There is a methodological limitation in the cell counting approach, as no distinction has been made between live and dead cells. It is recommended to use viability dyes such as Trypan Blue or Fluorescein Diacetate to differentiate between viable and non-viable cells.

• It is stated that two fields were counted, but it is not clear how many times the counting was repeated (three times to reduce human error). I recommend adding a sentence such as: Each sample was counted in triplicate to ensure accuracy and reproducibility.

• Ambiguity in "colony counts": If the term "colony" refers only to Microcystis colonies, it should be clearly stated, since some cyanobacteria do not form colonies.

• In this section, it is stated: "To determine cell density per mL, the average count was multiplied by the chamber’s dilution factor." However, it is unclear whether an actual dilution was performed on the samples or if this refers only to a calculation factor (such as the volume of each Neubauer chamber square).

• Although temperature and light intensity have been properly reported, the dissolved oxygen (DO) levels, oxygen concentration, ventilation intensity, and the type and rate of aeration are not specified. This information is essential for the reproducibility of the experiment and for accurately assessing the growth conditions.

• The sentence in line 90 may cause some confusion because the experiment involves three different concentrations considered as treatments, four types of discharge water, and one control. To improve clarity, it would be better to provide more details as follows: "In total, the experiment included four types of discharge water, each tested at three concentrations, plus one control, with replicates."

• It is recommended to specify the exact number of replicates, if applicable.

• Given the lack of prior studies, concentrations of 25%, 50%, and 100% were empirically selected to assess the growth of Microcystis aeruginosa and represent varying exposure levels to discharge waters. The design with doubling concentration ratios facilitates the identification of biological response thresholds. One limitation of the study is the absence of intermediate concentrations, such as 75%, which could have provided more precise information on the dose–response relationship and growth threshold points of the cyanobacterial species. Including such concentrations in future studies would help improve understanding of the gradual effects of discharge waters.

Results: The authors have made an effort to clearly explain the results; however, certain parts still require correction and improvement. For instance, the description of Figure 2 does not fully correspond to the data shown in the chart, and all supplementary tables are written in Spanish, which should be translated into English for consistency and clarity.

• There are typographical and grammatical errors from lines 238 to 260, which have been highlighted.

• In line 259, E6 was mistakenly written instead of E1 (tail), as the tail station corresponds to E1 and not E6. Therefore, the correct sentence should be: "Observed that the PTAR exit station (E2) presented the highest cell density (1.8 × 10⁶ cells/mL), followed by the tail station (E1) (5.3 × 10⁵ cells/mL) and Puerto Buga (E4) (5 × 10⁵ cells/mL)."

• There is a significant discrepancy between the cyanobacterial density values reported in the manuscript text and those presented in Figure 2. For instance, the manuscript states that the highest density observed during Campaign 2 (April) at Station E2 was 1.8 million cells/mL, whereas the corresponding value in the figure appears to be closer to 6 million cells/mL. Similarly, the text cites 1.5 million cells/mL as the overall maximum, which is inconsistent with the much higher values clearly depicted in the figure. These inconsistencies should be addressed by either correcting the textual data or clarifying whether the figure reflects updated results. Ensuring consistency between the text and the figures is essential for data transparency and accurate interpretation.

• The highest densities of potentially toxic cyanobacteria were observed during campaign 1 (February), with a peak value of approximately 6×10⁶ -7×10⁶ cells/mL at the PTAR exit station (E2). This campaign, corresponding to the dry season, showed markedly elevated cyanobacterial abundance compared to the other three campaigns. In contrast, campaign 2 (April) exhibited moderate densities, while campaigns 3 (July) and 4 (September) presented significantly lower concentrations, with minimal presence across most stations. However, this observation contradicts the description in the text, which incorrectly states that the highest densities occurred during campaign 2 (April).

• Please verify whether Figure 2 reflects updated data that have not been fully incorporated into the main text, or if a labelling or data transcription error occurred. Ensuring internal consistency between the figure and text is essential for data reliability. Additionally, could it be possible that this figure actually shows the total phytoplankton density across the six stations and four sampling periods, and that the caption for Figure 2 mistakenly refers to the density of potentially toxic cyanobacteria?

• Although February (the second month of the year) typically corresponds to winter in the Northern Hemisphere and summer in the Southern Hemisphere, it is important to note that countries located near the equator, such as Colombia, do not experience four distinct seasons. Instead, they generally have two main climatic periods: a dry season and a rainy season. Since Colombia is in the Northern Hemisphere but close to the equator, February usually falls within the dry season. It is therefore recommended that the authors mention this in the text to avoid potential confusion.

• In Fig. 3, the font on the x-axis (Sampling season) and the labels for the months (April, Feb, July, Sept) appear slightly tilted and small, which may make them difficult to read, especially in printed versions of the article. Additionally, the species names in the legend (Cyano, Aphanocapsa, M. flos-aquae, etc.) are somewhat crowded together. Increasing the spacing between them or using a bolder font could enhance readability. Finally, the y-axis labels in some panels (such as E3, E5, and E6) are quite small and compressed, which may reduce clarity.

• The axes in Figure 4 are very unclear and not suitable for publication. Please revise and improve their readability to ensure the figure meets publication standards.

• Line 317: Please add the word "and" before "orthophosphates."

• Line 325: The word "favored" is repeated twice; please delete one occurrence to avoid redundancy.

• There is an inconsistency between the numbering of treatments in the methodology section and the figure caption. In the methods, Treatment 1 corresponds to 100%, Treatment 2 to 50%, and Treatment 3 to 25%, whereas in the figure caption, C1 is labeled as 25%, C2 as 50%, and C3 as 100%. This mismatch may confuse readers when interpreting the results. It is recommended to standardize the treatment labels across the text and figures for clarity and consistency.

• There are grammatical issues in lines 338 and 339 that should be corrected.

• The supplementary materials have not been mentioned, and additionally, the supplementary tables are written in Spanish and need to be translated into English.

• Some stylistic errors in lines 376 to 398 need to be corrected.

• Some stylistic errors in lines 413 to 414 need to be corrected.

Discussion: The discussion is well-written, and the results are thoroughly compared with previous studies; however, some minor writing errors need to be corrected.

• All the writing errors in the discussion section have been highlighted. For example, reference 15 in line 458 was mistakenly italicized, there is no period after references 14 and 15 in line 447, and in line 432, it would be better to add "which" before "were consistent," among other issues.

Conclusion: The conclusion is well-written and only contains a few minor writing errors.

• In lines 512 and 513, a comma should be placed after the words system, discharges, and parameters.

• General Comment: The article addresses an interesting topic, and it is evident that the authors have invested significant time and effort in conducting the study. However, several issues need to be addressed. If these are properly corrected, the manuscript will be suitable for publication. I leave the final decision regarding the acceptance or rejection of the article at this stage to the discretion of the respected editor.

• A notable point in this study is the significant increase in the growth of the cyanobacterium Microcystis aeruginosa following the addition of water from tributary number 2 (T2) to the BG-11 culture medium. This indicates that T2 water contains specific nutrients and physicochemical characteristics that can optimize laboratory growth conditions for cyanobacteria. The most important of these features include a high concentration of ammonium nitrogen, elevated alkalinity and pH, and a suitable ratio of minerals such as calcium, magnesium, and phosphate. This unique composition provides a mineral-rich environment with ideal conditions that promote faster cyanobacterial growth. Identifying these factors and their targeted application to modify the BG-11 culture medium can lead to the development of more effective, reproducible, and cost-efficient cultivation media. An increased growth rate translates to higher biomass production in a shorter period — biomass that can serve as a valuable source for extracting lipids required in biofuel production, such as biodiesel, as well as bioactive compounds like phycocyanin and anticancer secondary metabolites. These achievements are especially significant when using non-toxic strains of M. aeruginosa that do not produce microcystins. Ultimately, such an approach can reduce cultivation time, save on the consumption of chemical components in the culture medium, and lower the overall costs of producing biotechnological products.

• The manuscript contains numerous language and writing issues that should have been thoroughly edited by the authors before submission. Additionally, there is a discrepancy between the box whisker plot presented in Figure 2 and its caption and explanation. It appears that the figure represents the total phytoplankton density, but the authors have mistakenly stated that it shows the density of harmful cyanobacteria. Furthermore, the supplementary tables are provided in Spanish, whereas they should have been translated into English before submission.

Reviewer #2: The abundance and proliferation of toxic cyanobacterial blooms have significant health and economic impacts. The authors have studied bloom dynamics in a lake important for recreation and human health. Overall the study is nicely presented and soundly designed. My only issue with the manuscript is that it could perhaps be better organised and presented, particularly the introduction and results section.

Also I have a small comment for Line 247-249:I am a but unclear about the terminology and ‘proportion’ of cyanobacteria. How is that calculated exactly? Does it mean that 89% of cells within the sample belong to cyanobacteria and the remaining 11% belong are other organisms? To make it more clear, could the authors simply refer to the cells/mL measured.

**Do you want your identity to be public for this peer review?** For information about this choice, including consent withdrawal, please see our Privacy Policy

Reviewer #1: No

Reviewer #2: No

---

## [Author Response · Author response to Decision Letter 1]

31 Oct 2025

Dear Editor:

We are sending you the paper entitled "Influence of environmental factors and tributaries on toxic cyanobacterial growth". We answered the questions of referees and revised according to the suggestions.

---

## [Decision Letter · Decision Letter 1]

15 Dec 2025

Influence of environmental factors and tributaries on toxic cyanobacterial growth

PONE-D-25-37791R1

Dear Dr. Natalia,

We’re pleased to inform you that your manuscript has been judged scientifically suitable for publication and will be formally accepted for publication once it meets all outstanding technical requirements.

Kind regards,

Barathan Balaji Prasath

Academic Editor

PLOS One

Additional Editor Comments (optional):

Reviewers' comments:

Reviewer's Responses to Questions

**Comments to the Author**

Reviewer #1: All comments have been addressed

2. Is the manuscript technically sound, and do the data support the conclusions?

Reviewer #1: Yes

3. Has the statistical analysis been performed appropriately and rigorously?

Reviewer #1: Yes

4. Have the authors made all data underlying the findings in their manuscript fully available?

Reviewer #1: Yes

5. Is the manuscript presented in an intelligible fashion and written in standard English?

Reviewer #1: Yes

Reviewer #1: Please be informed that all comments have been fully addressed, and the revised manuscript is now ready for acceptance. I respectfully leave the final decision to the esteemed editor.

**Do you want your identity to be public for this peer review?** For information about this choice, including consent withdrawal, please see our Privacy Policy

Reviewer #1: No

---

## [Editor Report · Acceptance letter]

PONE-D-25-37791R1

PLOS One

Dear Dr. Herrera,

I'm pleased to inform you that your manuscript has been deemed suitable for publication in PLOS One. Congratulations! Your manuscript is now being handed over to our production team.

Kind regards,

on behalf of

Dr. Barathan Balaji Prasath

Academic Editor

PLOS One